# Gradient Projection Memory for Continual Learning

**Gobinda Saha, Isha Garg & Kaushik Roy**
School of Electrical and Computer Engineering, Purdue University
`gsaha@purdue.edu, gargi@purdue.edu, kaushik@purdue.edu`

## Abstract

The ability to learn continually without forgetting the past tasks is a desired attribute for artificial learning systems. Existing approaches to enable such learning in artificial neural networks usually rely on network growth, importance based weight update or replay of old data from the memory. In contrast, we propose a novel approach where a neural network learns new tasks by taking gradient steps in the orthogonal direction to the gradient subspaces deemed important for the past tasks. We find the bases of these subspaces by analyzing network representations (activations) after learning each task with Singular Value Decomposition (SVD) in a single shot manner and store them in the memory as Gradient Projection Memory (GPM). With qualitative and quantitative analyses, we show that such orthogonal gradient descent induces minimum to no interference with the past tasks, thereby mitigates forgetting. We evaluate our algorithm on diverse image classification datasets with short and long sequences of tasks and report better or on-par performance compared to the state-of-the-art approaches[1].

## 1 Introduction

Humans exhibit remarkable ability in continual adaptation and learning new tasks throughout their lifetime while maintaining the knowledge gained from past experiences. In stark contrast, Artificial Neural Networks (ANNs) under such Continual Learning (CL) paradigm (Ring, 1998; Thrun & Mitchell, 1995; Lange et al., 2021) forget the information learned in the past tasks upon learning new ones. This phenomenon is known as 'Catastrophic Forgetting' or 'Catastrophic Interference' (Mccloskey & Cohen, 1989; Ratcliff, 1990). The problem is rooted in the general optimization methods (Goodfellow et al., 2016) that are being used to encode input data distribution into the parametric representation of the network during training. Upon exposure to a new task, gradient-based optimization methods, without any constraint, change the learned encoding to minimize the objective function with respect to the current data distribution. Such parametric updates lead to forgetting.

Given a fixed capacity network, one way to address this problem is to put constraints on the gradient updates so that task specific knowledge can be preserved. To this end, Kirkpatrick et al. (2017), Zenke et al. (2017), Aljundi et al. (2018), Serrà et al. (2018) add a penalty term to the objective function while optimizing for new task. Such term acts as a structural regularizer and dictates the degree of stability-plasticity of individual weights. Though these methods provide resource efficient solution to the catastrophic forgetting problem, their performance suffer while learning longer task sequence and when task identity is unavailable during inference.

Approaches (Lopez-Paz & Ranzato, 2017; Chaudhry et al., 2019a) that store episodic memories of old data essentially solve an optimization problem with 'explicit' constraints on the new gradient directions so that losses for the old task do not increase. In Chaudhry et al. (2019b) the performance of old task is retained by taking gradient steps in the average gradient direction obtained from the new data and memory samples. To minimize interference, Farajtabar et al. (2020) store gradient directions (instead of data) of the old tasks and optimize the network in the orthogonal directions to these gradients for the new task, whereas Zeng et al. (2018) update gradients orthogonal to the old input directions using projector matrices calculated iteratively during training. However, these methods either compromise data privacy by storing raw data or utilize resources poorly, which limits their scalability.

---

[1]Our code is available at `https://github.com/sahagobinda/GPM`

In this paper, we address the problem of catastrophic forgetting in a fixed capacity network when data from the old tasks are not available. To mitigate forgetting, our approach puts explicit constraints on the gradient directions that the optimizer can take. However, unlike contemporary methods, we neither store old gradient directions nor store old examples for generating reference directions. Instead we propose an approach that, after learning each task, partitions the entire gradient space of the weights into two orthogonal subspaces: Core Gradient Space (CGS) and Residual Gradient Space (RGS) (Saha et al., 2020). Leveraging the relationship between the input and the gradient spaces, we show how learned representations (activations) form the bases of these gradient subspaces in both fully-connected and convolutional networks. Using Singular Value Decomposition (SVD) on these activations, we show how to obtain the minimum set of bases of the CGS by which past knowledge is preserved and learnability for the new tasks is ensured. We store these bases in the memory which we define as Gradient Projection Memory (GPM). In our method, we propose to learn any new task by taking gradient steps in the orthogonal direction to the space (CGS) spanned by the GPM. Our analysis shows that such orthogonal gradient descent induces minimum to no interference with the old learning, and thus effective in alleviating catastrophic forgetting. We evaluate our approach in the context of image classification with miniImageNet, CIFAR-100, PMNIST and sequence of 5-Datasets on a variety of network architectures including ResNet. We compare our method with related state-of-the-art approaches and report comparable or better classification performance. Overall, we show that our method is memory efficient and scalable to complex dataset with longer task sequence while preserving data privacy.

## 2 RELATED WORKS

Approaches to continual learning for ANNs can be broadly divided into three categories. In this section we present a detailed discussion on the representative works from each category, highlighting their contributions and differences with our approach.

**Expansion-based methods:** Methods in this category overcome catastrophic forgetting by dedicating different subsets of network parameters to each task. With no constraint on network architecture, Progressive Neural Network (PGN) (Rusu et al., 2016) preserves old knowledge by freezing the base model and adding new sub-networks with lateral connections for each new task. Dynamically Expandable Networks (DEN) (Yoon et al., 2018) either retrains or expands the network by splitting/duplicating important units on new tasks, whereas Sarwar et al. (2020) grow the network to learn new tasks while sharing part of the base network. Li et al. (2019) with neural architecture search (NAS) find optimal network structures for each sequential task. RCL (Xu & Zhu, 2018) adaptively expands the network at each layer using reinforcement learning, whereas APD (Yoon et al., 2020) additively decomposes the parameters into shared and task specific parameters to minimize the increase in the network complexity. In contrast, our method avoids network growth or expensive NAS operations and performs sequential learning within a fixed network architecture.

**Regularization-based methods:** These methods attempt to overcome forgetting in fixed capacity model through structural regularization which penalizes major changes in the parameters that were important for the previous tasks. Elastic Weight Consolidation (EWC) (Kirkpatrick et al., 2017) computes such importance from diagonal of Fisher information matrix after training, whereas Zenke et al. (2017) compute them during training based on loss sensitivity with respect to the parameters. Additionally, Aljundi et al. (2018) compute importance from sensitivity of model outputs to the inputs. Other methods, such as PackNet (Mallya & Lazebnik, 2018) uses iterative pruning to fully restrict gradient updates on important weights via binary mask, whereas HAT (Serrà et al., 2018) identifies important neurons by learning attention masks that control gradient propagation in the individual parameters. Saha et al. (2020) using a PCA based pruning on activations (Garg et al., 2020) partition the parametric space of the weights (filters) into core and residual (filter) spaces after learning each task. The past knowledge is preserved in the frozen core space, whereas the residual space is updated when learning the next task. In contrast to these methods, we do not ascribe importance to or restrict the gradients of any individual parameters or filters. Rather we put constraints on the 'direction' of gradient descent.

**Memory-based methods:** Methods under this class mitigate forgetting by either storing a subset of (raw) examples from the past tasks in the memory for rehearsal (Robins, 1995; Rebuffi et al., 2017; Lopez-Paz & Ranzato, 2017; Chaudhry et al., 2019a;b; Riemer et al., 2019) or synthesizing old data from generative models to perform pseudo-rehearsal (Shin et al., 2017). For instance,

Gradient Episodic Memory (GEM) (Lopez-Paz & Ranzato, 2017) avoids interference with previous task by projecting the new gradients in the feasible region outlined by previous task gradients calculated from the samples of episodic memory. Averaged-GEM (A-GEM) (Chaudhry et al., 2019a) simplified this optimization problem to projection in one direction estimated by randomly selected samples from the memory. Guo et al. (2020) propose a unified view of episodic memory-based CL methods, that include GEM and A-GEM and improves performance over these methods utilizing loss-balancing update rule. Additionally, Experience Replay (ER) (Chaudhry et al., 2019b) and Meta-Experience Replay (MER) (Riemer et al., 2019) mitigate forgetting in online CL setup by jointly training on the samples from new tasks and episodic memory. All these methods, however, rely on the access to old data which might not be possible when users have concern over data privacy. Like all the memory-based methods we also use a storage unit which we call GPM. However, we do not save any raw data in GPM, thus satisfy data privacy criterion.

Our method is closely related to recently proposed Orthogonal Gradient Descent (OGD) (Farajtabar et al., 2020) and Orthogonal Weight Modulation (OWM) (Zeng et al., 2018). OGD stores a set of gradient directions in the memory for each task and minimizes catastrophic forgetting by taking gradient steps in the orthogonal directions for new tasks. In contrast to OGD, we compute and store the bases of core gradient space from network representations (activations) which reduces the memory requirement by orders of magnitude. Moreover, OGD is shown to work under locality assumption for small learning rates which limits its scalability in learning longer task sequences with complex dataset. Since our method does not use gradient directions (like OGD) to describe the core gradient spaces, we do not need to obey such assumptions, thus can use higher learning rates. On the other hand, OWM reduces forgetting by modifying the weights of the network in the orthogonal to the input directions of the past tasks. This is achieved by multiplying new gradients with projector matrices. These matrices are computed from the stored past projectors and the inputs with recursive least square (RLS) method at each training step. However, such an iterative method not only slows down the training process but also shows limited scalability in end-to-end task learning with modern network architectures. Like OWM, we aim to encode new learning in the orthogonal to the old input directions. In contrast to iterative projector computation in OWM, we identify a low-dimensional subspace in the gradient space analyzing the learned representations with SVD in one-shot manner at the end of each task. We store the bases of these subspaces in GPM and learn new tasks in the orthogonal to these spaces to protect old knowledge. We quantitatively show that our method is memory efficient, fast and scalable to deeper networks for complex long sequence of tasks.

## 3 NOTATIONS AND BACKGROUND

In this section, we introduce the notations used throughout the paper and give a brief overview of SVD for matrix approximation. In section 4, we establish the relationship between input and gradient spaces. In section 5 we show the steps of our algorithm that leverage such relationship.

**Continual Learning:** We consider supervised learning setup where $T$ tasks are learned sequentially. Each task has a task descriptor, $\tau \in \{1, 2, ..., T\}$ with a corresponding dataset, $\mathbb{D}_\tau = \{(\boldsymbol{x}_{i,\tau}, \boldsymbol{y}_{i,\tau})_{i=1}^{n_\tau}\}$ having $n_\tau$ example pairs. Let's consider an $L$ layer neural network where at each layer network computes the following function for task $\tau$ :

$$\boldsymbol{x}_{i,\tau}^{l+1} = \sigma(f(\boldsymbol{W}_\tau^l, \boldsymbol{x}_{i,\tau}^l)). \tag{1}$$

Here, $l = 1, ...L$, $\sigma(.)$ is a non-linear function and $f(.,.)$ is a linear function. We will use vector notation for input ($\boldsymbol{x}_{i,\tau}$) in fully connected layers and matrix notation for input ($\boldsymbol{X}_{i,\tau}$) in convolutional layers. At the first layer, $\boldsymbol{x}_{i,\tau}^1 = \boldsymbol{x}_{i,\tau}$ represents the raw input data from task $\tau$, whereas in the subsequent layers we define $\boldsymbol{x}_{i,\tau}^l$ as the **representation of input** $\boldsymbol{x}_{i,\tau}$ at layer $l$. Set of parameters of the network is defined by, $\mathbb{W}_\tau = \{(\boldsymbol{W}_\tau^l)_{l=1}^L\}$, where $\mathbb{W}_0$ denotes set of parameters at initialization.

**Matrix approximation with SVD:** SVD can be used to factorize a rectangular matrix, $\boldsymbol{A} = \boldsymbol{U}\boldsymbol{\Sigma}\boldsymbol{V}^T \in \mathbb{R}^{m \times n}$ into the product of three matrices, where $\boldsymbol{U} \in \mathbb{R}^{m \times m}$ and $\boldsymbol{V} \in \mathbb{R}^{n \times n}$ are orthogonal, and $\boldsymbol{\Sigma}$ contains the sorted singular values along its main diagonal (Deisenroth et al., 2020). If the rank of the matrix is $r$ ($r \leq \min(m, n)$), $\boldsymbol{A}$ can be expressed as $\boldsymbol{A} = \sum_{i=1}^r \sigma_i \boldsymbol{u}_i \boldsymbol{v}_i^T$, where $\boldsymbol{u}_i \in \boldsymbol{U}$ and $\boldsymbol{v}_i \in \boldsymbol{V}$ are left and right singular vectors and $\sigma_i \in diag(\boldsymbol{\Sigma})$ are singular values. Also, $k$-rank approximation to this matrix can be expressed as, $\boldsymbol{A}_k = \sum_{i=1}^k \sigma_i \boldsymbol{u}_i \boldsymbol{v}_i^T$, where $k \leq r$ and its value can be chosen by the smallest $k$ that satisfies $||\boldsymbol{A}_k||_F^2 \geq \epsilon_{th}||\boldsymbol{A}||_F^2$. Here, $||.||_F$ is the Frobenius norm of the matrix and $\epsilon_{th}$ ($0 < \epsilon_{th} \leq 1$) is the threshold hyperparameter.

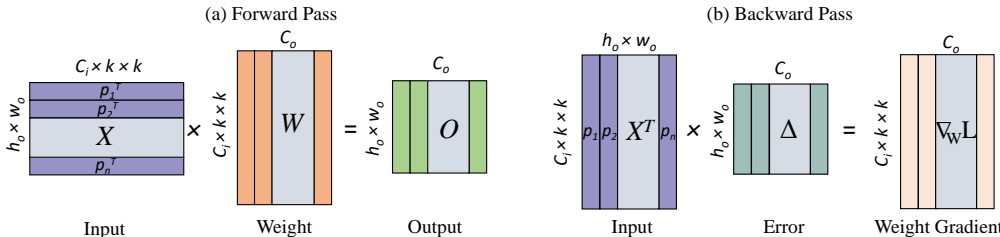

Figure 1: Illustration of convolution operation in matrix multiplication format during (a) Forward Pass and (b) Backward Pass.

## 4 INPUT AND GRADIENT SPACES

Our algorithm leverages the fact that stochastic gradient descent (SGD) updates lie in the span of input data points (Zhang et al., 2017). In the following subsections we will establish this relationship for both fully connected and convolutional layers. The analysis presented in this section is generally applicable to any layer of the network for any task, and hence we drop the task and layer identifiers.

### 4.1 FULLY CONNECTED LAYER

Let's consider a single layer linear neural network in supervised learning setup where each (input, label) training data pair comes from a training dataset, $\mathbb{D}$. Let, $\boldsymbol{x} \in \mathbb{R}^n$ is the input vector, $\boldsymbol{y} \in \mathbb{R}^m$ is the label vector in the dataset and $\boldsymbol{W} \in \mathbb{R}^{m \times n}$ are the parameters (weights) of the network. The network is trained by minimizing the following mean-squared error loss function

$$L = \frac{1}{2} ||\boldsymbol{W}\boldsymbol{x} - \boldsymbol{y}||_2^2. \tag{2}$$

We can express gradient of this loss with respect to weights as

$$\nabla_{\boldsymbol{W}} L = (\boldsymbol{W}\boldsymbol{x} - \boldsymbol{y})\boldsymbol{x}^T = \boldsymbol{\delta}\boldsymbol{x}^T, \tag{3}$$

where $\boldsymbol{\delta} \in \mathbb{R}^m$ is the error vector. Thus, **the gradient update will lie in the span of input ($\boldsymbol{x}$)**, where elements in $\boldsymbol{\delta}$ scale the magnitude of $\boldsymbol{x}$ by different factors. Here, we have considered per-example loss (batch size of 1) for simplicity. However, this relation also holds for mini-batch setting (see appendix B.1). The input-gradient relation in equation 3 is generically applicable to any fully connected layer of a neural network where $\boldsymbol{x}$ is the input to that layer and $\boldsymbol{\delta}$ is the error coming from the next layer. Moreover, this equation also holds for network with non-linear units (*e.g.* ReLU) and cross-entropy losses except the calculation of $\boldsymbol{\delta}$ will be different.

### 4.2 CONVOLUTIONAL LAYER

Filters in a convolutional (Conv) layer operate in a different way on the inputs than the weights in a fully connected (FC) layer. Let's consider a Conv layer with the input tensor $\mathcal{X} \in \mathbb{R}^{C_i \times h_i \times w_i}$ and filters $\mathcal{W} \in \mathbb{R}^{C_o \times C_i \times k \times k}$. Their convolution $\langle \mathcal{X}, \mathcal{W}, * \rangle$ produces output feature map, $\mathcal{O} \in \mathbb{R}^{C_o \times h_o \times w_o}$ (Liu et al., 2018). Here, $C_i$ ($C_o$) denotes the number of input (output) channels of the Conv layer, $h_i$, $w_i$ ($h_o$, $w_o$) denote the height and width of the input (output) feature maps and $k$ is the kernel size of the filters. As shown in Figure 1(a), if $\mathcal{X}$ is reshaped into a $(h_o \times w_o) \times (C_i \times k \times k)$ matrix, $\boldsymbol{X}$ and $\mathcal{W}$ is reshaped into a $(C_i \times k \times k) \times C_o$ matrix, $\boldsymbol{W}$, then the convolution can be expressed as matrix multiplication between $\boldsymbol{X}$ and $\boldsymbol{W}$ as $\boldsymbol{O} = \boldsymbol{X}\boldsymbol{W}$, where $\boldsymbol{O} \in \mathbb{R}^{(h_o \times w_o) \times C_o}$. Each row of $\boldsymbol{X}$ contains an input patch vector, $\boldsymbol{p}_j \in \mathbb{R}^{(C_i \times k \times k) \times 1}$, where $j = 1, 2..., n$ ($n = h_o * w_o$).

Formulation of convolution in terms of matrix multiplication provides an intuitive picture of the gradient computation during backpropagation. Similar to the FC layer case, in Conv layer, during backward pass an error matrix $\boldsymbol{\Delta}$ of size $(h_0 \times w_0) \times C_o$ (same size as $\boldsymbol{O}$) is obtained from the next layer. As shown in Figure 1(b), the gradient of loss with respect to filter weights is calculated by

$$\nabla_{\boldsymbol{W}} L = \boldsymbol{X}^T \boldsymbol{\Delta}, \tag{4}$$

where, $\nabla_{\boldsymbol{W}} L$ is of shape $(C_i \times k \times k) \times C_o$ (same size as $\boldsymbol{W}$). Since, columns of $\boldsymbol{X}^T$ are the input patch vectors ($\boldsymbol{p}$), **the gradient updates of the convolutional filters will lie in the space spanned by these patch vectors**.

## 5 CONTINUAL LEARNING WITH GRADIENT PROJECTION MEMORY (GPM)

In this section, we describe our continual learning algorithm which leverages the relationship between gradient and input spaces to identify the core gradient spaces of the past tasks. We show how gradient descent orthogonal to these spaces enable us to learn continually without forgetting.

**Learning Task 1:** We learn the first task ($\tau = 1$) using dataset, $\mathbb{D}_1$ without imposing any constraint on parameter updates. At the end of Task 1, we obtain a learned set of parameters $\mathbb{W}_1$. To preserve the knowledge of the learned task, we impose constraints on the direction of gradient updates for the next tasks. To do so, we partition the entire gradient space into two (orthogonal) subspaces: Core Gradient Space (CGS) and Residual Gradient Space (RGS), such that gradient steps along CGS induce high interference on the learned tasks whereas gradient steps along RGS have minimum to no interference. We aim to find and store the bases of the CGS and take gradient steps orthogonal to the CGS for the next task. In our formulation, each layer has its own CGS.

To find the bases, after learning Task 1 , for each layer we construct a **representation matrix**, $\boldsymbol{R}_1^l = [\boldsymbol{x}_{1,1}^l, \boldsymbol{x}_{2,1}^l, ..., \boldsymbol{x}_{n_s,1}^l]$ (for Conv layers $\boldsymbol{R}_1^l = [(\boldsymbol{X}_{1,1}^l)^T, (\boldsymbol{X}_{2,1}^l)^T, ..., (\boldsymbol{X}_{n_s,1}^l)^T]$ ) concatenating $n_s$ representations along the column obtained from forward pass of $n_s$ random samples from the current training dataset through the network. Next, we perform SVD on $\boldsymbol{R}_1^l = \boldsymbol{U}_1^l \boldsymbol{\Sigma}_1^l (\boldsymbol{V}_1^l)^T$ followed by its $k$-rank approximation $(\boldsymbol{R}_1^l)_k$ according to the following **criteria** for the given threshold, $\epsilon_{th}^l$ :

$$||(\boldsymbol{R}_1^l)_k||_F^2 \geq \epsilon_{th}^l ||\boldsymbol{R}_1^l||_F^2. \tag{5}$$

We define the space, $S^l = span\{\boldsymbol{u}_{1,1}^l, \boldsymbol{u}_{2,1}^l, ..., \boldsymbol{u}_{k,1}^l\}$, spanned by the first $k$ vectors in $U_1^l$ as the **space of significant representation** for task 1 at layer $l$ since it contains all the directions with highest singular values in the representation. For the next task, we aim to take gradient steps in a way that the correlation between this task specific significant representation and the weights in each layer is preserved. Since, inputs span the space of gradient descent (section 4), the bases of $S^l$ will span a subspace in the gradient space which we define as the Core Gradient space (CGS). Thus gradient descent along CGS will cause maximum change in the input-weight correlation whereas gradient steps in the orthogonal directions to CGS (space of low representational significance) will induce very small to no interference to the old tasks. We define this subspace orthogonal to CGS as Residual Gradient space (RGS). We save the bases of the CGS in the memory, $\mathcal{M} = \{(\boldsymbol{M}^l)_{l=1}^L\}$, where $\boldsymbol{M}^l = [\boldsymbol{u}_{1,1}^l, \boldsymbol{u}_{2,1}^l, ..., \boldsymbol{u}_{k,1}^l]$. We define this memory as Gradient Projection Memory (GPM).

**Learning Task 2 to T:** We learn task 2 with the examples from dataset $\mathbb{D}_2$ only. Before taking gradient step, bases of the CGS are retrieved from GPM. New gradients ($\nabla_{\boldsymbol{W}_2^l} L_2$) are first projected onto the CGS and then projected components are subtracted out from the new gradient so that remaining gradient components lie in the space orthogonal to CGS. Gradients are updated as

$$\text{FC Layer:} \quad \nabla_{\boldsymbol{W}_2^l} L_2 = \nabla_{\boldsymbol{W}_2^l} L_2 - (\nabla_{\boldsymbol{W}_2^l} L_2)\boldsymbol{M}^l(\boldsymbol{M}^l)^T, \tag{6}$$

$$\text{Conv Layer:} \quad \nabla_{\boldsymbol{W}_2^l} L_2 = \nabla_{\boldsymbol{W}_2^l} L_2 - \boldsymbol{M}^l(\boldsymbol{M}^l)^T(\nabla_{\boldsymbol{W}_2^l} L_2). \tag{7}$$

At the end of the task 2 training, we update the GPM with new task-specific bases (of CGS). To obtain such bases, we construct $\boldsymbol{R}_2^l = [\boldsymbol{x}_{1,2}^l, \boldsymbol{x}_{2,2}^l, ..., \boldsymbol{x}_{n_s,2}^l]$ using data from task 2 only. However, before performing SVD and subsequent $k$-rank approximation, from $\boldsymbol{R}_2^l$ we eliminate the common directions (bases) that are already present in the GPM so that newly added bases are unique and orthogonal to the existing bases in the memory. To do so, we perform the following step :

$$\hat{\boldsymbol{R}}_2^l = \boldsymbol{R}_2^l - \boldsymbol{M}^l(\boldsymbol{M}^l)^T(\boldsymbol{R}_2^l) = \boldsymbol{R}_2^l - \boldsymbol{R}_{2,Proj}^l. \tag{8}$$

Afterwards, SVD is performed on $\hat{\boldsymbol{R}}_2^l$ ($= \hat{\boldsymbol{U}}_2^l \hat{\boldsymbol{\Sigma}}_2^l (\hat{\boldsymbol{V}}_2^l)^T$) and $k$ new orthogonal bases are chosen for minimum value of $k$ satisfying the following **criteria** for the given threshold, $\epsilon_{th}^l$:

$$||\boldsymbol{R}_{2,proj}^l||_F^2 + ||(\hat{\boldsymbol{R}}_2^l)_k||_F^2 \geq \epsilon_{th}^l ||\boldsymbol{R}_2^l||_F^2. \tag{9}$$

GPM is updated by adding new bases as $\boldsymbol{M}^l = [\boldsymbol{M}^l, \hat{\boldsymbol{u}}_{1,2}^l, ..., \hat{\boldsymbol{u}}_{k,2}^l]$. Thus after learning each new task, CGS grows and RGS becomes smaller, where maximum size of $\boldsymbol{M}^l$ (hence the dimension of the gradient bases) is fixed by the choice of initial network architecture. Once the GPM update is complete we move on to the next task and repeat the same procedure that we followed for task 2. The pseudo-code of the algorithm is given in Algorithm 1 in the appendix.

## 6    EXPERIMENTAL SETUP

**Datasets:** We evaluate our continual learning algorithm on **Permuted MNIST** (PMNIST) (Lecun et al., 1998), **10-Split CIFAR-100** (Krizhevsky, 2009), **20-Spilt miniImageNet** (Vinyals et al., 2016) and sequence of **5-Datasets** (Ebrahimi et al., 2020b). The PMNIST dataset is a variant of MNIST dataset where each task is considered as a random permutation of the original MNIST pixels. For PMNIST, we create 10 sequential tasks using different permutations where each task has 10 classes (Ebrahimi et al., 2020a). The 10-Split CIFAR-100 is constructed by splitting 100 classes of CIFAR-100 into 10 tasks with 10 classes per task. Whereas, 20-Spilt miniImageNet, used in (Chaudhry et al., 2019a), is constructed by splitting 100 classes of miniImageNet into 20 sequential tasks where each task has 5 classes. Finally, we use a sequence of 5-Datasets including CIFAR-10, MNIST, SVHN (Netzer et al., 2011), notMNIST (Bulatov, 2011) and Fashion MNIST (Xiao et al., 2017), where classification on each dataset is considered as a task. In our experiments we do not use any data augmentation. The dataset statistics are given in Table 4 & 5 in the appendix.

**Network Architecture:** We use fully-connected network with two hidden layer of 100 units each for PMNIST following Lopez-Paz & Ranzato (2017). For experiments with split CIFAR-100 we use a 5-layer AlexNet similar to  Serrà et al. (2018). For split miniImageNet and 5-Datasets, similar to Chaudhry et al. (2019b), we use a reduced ResNet18 architecture. No bias units are used and batch normalization parameters are learned for the first task and shared with all the other tasks (following Mallya & Lazebnik (2018)). Details on architectures are given in the appendix section C.2. For permuted MNIST, we evaluate and compare our algorithm in 'single-head' setting (Hsu et al., 2018; Farquhar & Gal, 2018) where all tasks share the final classifier layer and inference is performed without task hint. For all other experiments, we evaluate our algorithm in 'muti-head' setting, where each task has a separate classifier on which no gradient constraint is imposed during learning.

**Baselines:** We compare our method with state-of-the art approaches from both memory based and regularization based methods that consider sequential task learning in fixed network architecture. From memory based approach, we compare with Experience Replay with reservoir sampling (ER_Res) (Chaudhry et al., 2019b), Gradient Episodic Memory (GEM) (Lopez-Paz & Ranzato, 2017), Averaged GEM (A-GEM) (Chaudhry et al., 2019a), Orthogonal Gradient Descent (OGD) (Farajtabar et al., 2020) and Orthogonal Weight Modulation (OWM) (Zeng et al., 2018). Moreover, we compare with sate-of-the-art HAT (Serrà et al., 2018) baseline and Elastic Weight Consolidation (EWC) (Kirkpatrick et al., 2017) from regularization based methods. Additionally, we add 'multitask' baseline where all the tasks are learned jointly using the entire dataset at once in a single network. Multitask is not a continual learning strategy but will serve as upper bound on average accuracy on all tasks. Details on the implementation along with the hyperparameters considered for each of these baselines are provided in section C.4 and Table 6 in the appendix.

**Training Details:** We train all the models with plain stochastic gradient descent (SGD). For each task in PMNIST and split miniImageNet we train the network for 5 and 10 epochs respectively with batch size of 10. In Split CIFAR-100 and 5-Datasets experiments, we train each task for maximum of 200 and 100 epochs respectively with the early termination strategy based on the validation loss as proposed in Serrà et al. (2018). For both datasets, batch size is set to 64. For GEM, A-GEM and ER_Res the episodic memory size is chosen to be approximately the same size as the maximum GPM size (GPM_Max). Calculation of GPM size is given in Table 7 in the appendix. Moreover, selection of the threshold values ($\epsilon_{th}$) in our method is discussed in section C.5 in the appendix.

**Performance Metrics:** To evaluate the classification performance, we use the **ACC** metric, which is the average test classification accuracy of all tasks. To measure the forgetting we report backward transfer, **BWT** which indicates the influence of new learning on the past knowledge. For instance, negative BWT indicates (catastrophic) forgetting. Formally, ACC and BWT are defined as:

$$\text{ACC} = \frac{1}{T}\sum_{i=1}^{T} R_{T,i}, \quad \text{BWT} = \frac{1}{T-1}\sum_{i=1}^{T-1} R_{T,i} - R_{i,i}. \tag{10}$$

Here, $T$ is the total number of sequential tasks and $R_{T,i}$ is the accuracy of the model on $i^{th}$ task after learning the $T^{th}$ task sequentially (Lopez-Paz & Ranzato, 2017).

## 7    RESULTS AND DISCUSSIONS

**Single-head inference with PMNIST:** First, we evaluate our algorithm in single-head setup for 10 sequential PMNIST tasks. In this setup task hint is not necessary. As HAT cannot perform infer-

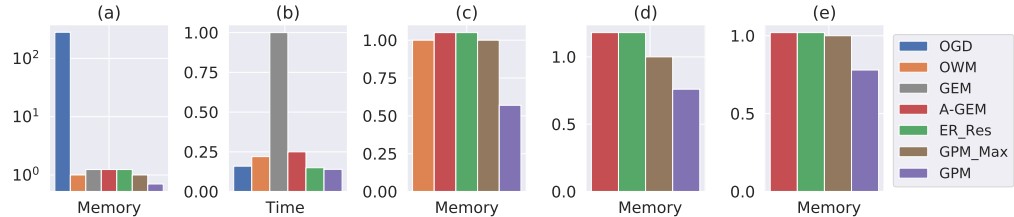

Figure 2: (a) Memory utilization and (b) per epoch training time for PMNIST tasks for different methods. Memory utilization for different approaches for (c) CIFAR-100, (d) miniImageNet and (e) 5-Datasets tasks. For memory, size of GPM_Max and for time, method with highest complexity is used as references (value of 1). All the other methods are reported relative to these references.

Table 1: Continual learning on different datasets. Methods that do not adhere to CL setup is indicated by (*). All the results are (re) produced by us and averaged over 5 runs. Standard deviations are reported in Table 8 and 9 in the appendix.

<table>
<tr><td colspan="3">(a)</td><td colspan="7">(b)</td></tr>
<tr><td></td><td colspan="2">PMNIST</td><td></td><td colspan="2">CIFAR-100</td><td colspan="2">miniImageNet</td><td colspan="2">5-Datasets</td></tr>
<tr><td>Methods</td><td>ACC (%)</td><td>BWT</td><td>Methods</td><td>ACC (%)</td><td>BWT</td><td>ACC (%)</td><td>BWT</td><td>ACC (%)</td><td>BWT</td></tr>
<tr><td>OGD</td><td>82.56</td><td>- 0.14</td><td>OWM</td><td>50.94</td><td>- 0.30</td><td>-</td><td>-</td><td>-</td><td>-</td></tr>
<tr><td>OWM</td><td>90.71</td><td>- 0.01</td><td>EWC</td><td>68.80</td><td>- 0.02</td><td>52.01</td><td>- 0.12</td><td>88.64</td><td>- 0.04</td></tr>
<tr><td>GEM</td><td>83.38</td><td>- 0.15</td><td>HAT</td><td>72.06</td><td>- 0.00</td><td>59.78</td><td>- 0.03</td><td>**91.32**</td><td>- 0.01</td></tr>
<tr><td>A-GEM</td><td>83.56</td><td>- 0.14</td><td>A-GEM</td><td>63.98</td><td>- 0.15</td><td>57.24</td><td>- 0.12</td><td>84.04</td><td>-0.12</td></tr>
<tr><td>ER_Res</td><td>87.24</td><td>- 0.11</td><td>ER_Res</td><td>71.73</td><td>- 0.06</td><td>58.94</td><td>- 0.07</td><td>88.31</td><td>- 0.04</td></tr>
<tr><td>EWC</td><td>89.97</td><td>- 0.04</td><td>GPM (ours)</td><td>**72.48**</td><td>- 0.00</td><td>**60.41**</td><td>- 0.00</td><td>91.22</td><td>- 0.01</td></tr>
<tr><td>GPM (ours)</td><td>**93.91**</td><td>-0.03</td><td></td><td></td><td></td><td></td><td></td><td></td><td></td></tr>
<tr><td>Multitask*</td><td>96.70</td><td>-</td><td>Multitask*</td><td>79.58</td><td>-</td><td>69.46</td><td>-</td><td>91.54</td><td>-</td></tr>
</table>

ence without task hint, it is not included in the comparison. Since network size is very small (0.1M parameters) with $87\%$ parameters in the first layer, we choose threshold value ($\epsilon_{th}$) of $0.95$ for that layer and $0.99$ for the other layers to ensure better learnability. From the results, shown in Table 1(a), we observe that our method (GPM) achieves best average accuracy ($93.91 \pm 0.16\%$). In addition, we achieve least amount of forgetting, except OWM, which essentially trades off accuracy to minimize forgetting. Figure 2(a) compares the memory utilization of all the memory-based approaches. While OWM, GEM, A-GEM and ER_Res use memory of size of GPM_Max, we obtain better performance by using only $69\%$ of the GPM_Max. Moreover, compared to OGD, we use about $400$ times lower memory and achieve $\sim 10\%$ better accuracy. In Figure 2(b), we compare the per epoch training time of different memory based methods and found our method to be the fastest primarily due to the precomputation of the reference gradient bases (of CGS). Additionally, in single-epoch setting (Lopez-Paz & Ranzato, 2017), as shown in Table 8 in the appendix, we obtain best average accuracy ($91.74 \pm 0.15\%$), which demonstrates the potential for our algorithm in online CL setup.

**Split CIFAR-100:** Next, we switch to multi-head setup which enables us to compare with strong baselines such as HAT. For ten split CIFAR-100 tasks, as shown in Table 1(b), we outperform all the memory based approaches while using $45\%$ less memory (Figure 2(c)). We also outperform EWC and our accuracy is marginally better than HAT while achieving zero forgetting. Also, we obtain $\sim 20\%$ better accuracy than OWM, which have high forgetting (BWT=$-0.30$) thus demonstrating its limited scalability to convolutional architectures.

**Split miniImageNet:** With this experiment, we test the scalability of our algorithm to deeper network (ResNet18) for long task sequence from miniImageNet dataset. The average accuracies for different methods after learning 20 sequential tasks are given in Table 1(b). Again, in this case we outperform A-GEM, ER_Res and EWC using $76\%$ of the GPM_Max (Figure 2(d)). Also, we achieve marginally better accuracy than HAT, however unlike HAT (and other methods) we completely avoid forgetting (BWT=0.00). Moreover, compared other methods sequential learning in our method is more stable, which means accuracy of the past tasks have minimum to no degradation over the course of learning (shown for task 1 accuracy in Figure 4 in the appendix).

**5-Datasets:** Next, we validate our approach on learning across diverse datasets, where classification on each dataset is treated as one task. Even in this challenging setting, as shown in in Table 1(b),

Table 2: Total wall-clock training time measured on a single GPU after learning all the tasks.

| (a) | | | (b) | | | |
|---|---|---|---|---|---|---|
| | **Training Time [s]** | | | **Training Time [s]** | | |
| **Methods** | PMNIST | | **Methods** | CIFAR-100 | miniImageNet | 5-Datasets |
| OGD | 1658 | | OWM | 1856 | - | - |
| OWM | 396 | | EWC | 1352 | 4138 | 7613 |
| GEM | 1639 | | HAT | 1248 | 3077 | 7246 |
| A-GEM | 445 | | A-GEM | 2678 | 6069 | 12077 |
| ER_Res | 259 | | ER_Res | 1147 | **2775** | 7015 |
| EWC | 645 | | GPM (ours) | **770** | 3387 | **5008** |
| GPM (ours) | **245** | | | | | |

Table 3: Continual learning of 20-task from CIFAR-100 Superclass dataset. (†) denotes the result reported from APD. (*) indicates the methods that do not adhere to CL setup. Single-task learning (STL), where a separate network in trained for each task, serves as an upper bound on accuracy.

| | **Methods** | | | | | |
|---|---|---|---|---|---|---|
| Metric | STL†* | PGN† | DEN† | RCL† | APD† | GPM (ours) |
| ACC (%) | 61.00 | 50.76 | 51.10 | 51.99 | 56.81 | **57.72** |
| Capacity (%) | 2000 | 271 | 191 | 184 | 130 | **100** |

we achieve better accuracy ($91.22 \pm 0.20\%$) then A-GEM, ER_Res and EWC utilizing 78% of the GPM_Max (Figure 2(e)). Though, HAT performs marginally better than our method, both HAT and we achieve the lowest BWT (-0.01). In this experiment, we have used tasks that are less related to each other. After learning 5 such tasks 78% of the gradient space is already constrained. Which implies, if the tasks are less or non-similar, GPM will get populated faster and reach to its maximum capacity after which no new learning will be possible. Since, we use a fixed capacity network and the size of GPM is determined by the network architecture, the ability of learning sequences of hundreds of such tasks with our method will be limited by the chosen network capacity.

**Training Time.** Table 2 shows the total training time for all the sequential tasks for different algorithms. This includes time spent for memory management for the memory-based methods, fisher importance calculation for EWC and learning activation masks for HAT. Details of time measurement are given in appendix section C.6. For PMNIST, CIFAR-100, and 5-dataset tasks our algorithm trains faster than all the other baselines while spending only 0.2%, 3% and 6% of its total training time in GPM update (using SVD) respectively. Since each miniImageNet tasks are trained for only 10 epochs, our method have relatively higher overhead (30% of the total time) due to GPM update, thus runs a bit slower than the fastest ER_Res. Overall, our formulation uses GPM bases and projection matrices of reasonable dimensions (see appendix section C.8); precomputation of which at the start of each task leads to fast per-epoch training. This gain in time essentially compensates for the extra time required for the GPM update, which is done only once per task, enabling fast training.

**Comparison with Expansion-based methods.** To compare our method with the state-of-the-art expansion based methods we perform experiment with 20-task CIFAR-100 Superclass dataset (Yoon et al., 2020). In this experiment, each task contains 5 different but semantically related classes from CIFAR-100 dataset. Similar to APD, here we use the LeNet-5 architecture. Details of architecture and training setup are given in appendix section C.3. Results are shown in Table 3 where ACC represents average accuracy over 5 different task sequences (used in APD) and Capacity denotes percentage of network capacity used with respect to the original network. We outperform all the CL methods achieving best average accuracy ($57.72 \pm 0.37\%$) with BWT of -0.01 using the smallest network. For instance, we outperform RCL and APD utilizing 84% and 30% fewer network parameters respectively, which shows that our method induces more sharing between tasks.

Overall, we outperform the memory-based methods with less memory utilization, achieve better accuracy than expansion-based methods using smaller network, and obtain better or on-par performance compared to HAT in the given experimental setups. However, in the class-incremental learning setup (Rebuffi et al., 2017), method (Kamra et al., 2017) that uses data replay achieves better performance than GPM (see experiment in appendix section D.3). In this setup, we believe a subset of old data replay either from storage or via generation is inevitable for attaining better per-

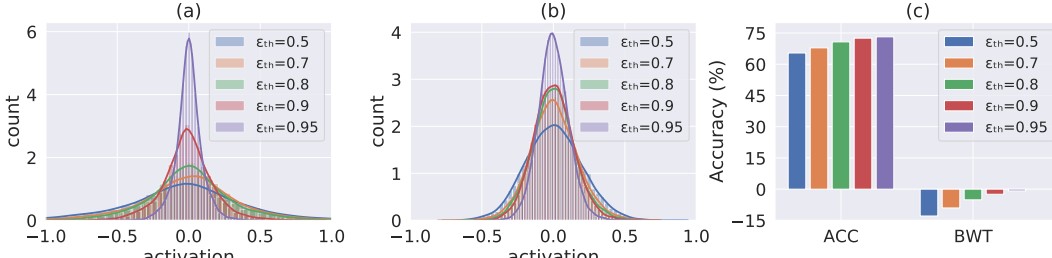

Figure 3: Histograms of interference activations as a function of threshold, ($\epsilon_{th}$) at (a) Conv layer 2 (b) FC layer 2 for split CIFAR-100 tasks. (c) Impact of $\epsilon_{th}$ on ACC (%) and BWT(%). With increasing value of $\epsilon_{th}$, spread of interference reduces, which improves accuracy and reduces forgetting.

formance with minimal forgetting (Rajasegaran et al., 2019). In that quest, a hybrid approach such as combining GPM with small data replay would be an interesting direction for future exploration.

**Controlling Forgetting:** Finally, we discuss the factors that implicitly or explicitly control the amount of forgetting in our algorithm. As discussed in section 5, we propose to minimize interference by taking gradient steps orthogonal to the CGS, where CGS bases are computed such that space of significant representations of the past tasks can be well approximated by these bases. The degree of this approximation is controlled by the threshold hyperparameter, $\epsilon_{th}$ (through equation 5, 9). For instance, a low value of $\epsilon_{th}$ (closer to 0) would allow the optimizer to change the weights along the directions where past data has higher representational significance, thereby significantly altering the past input-weight correlation inducing (catastrophic) interference. On the other hand, a high value of $\epsilon_{th}$ (closer to 1) would preserve such correlation, however learnability of the new task might suffer due to high volume of constraints in the gradient space. Therefore, in our continual learning algorithm, $\epsilon_{th}$ mediates the stability-plasticity dilemma. To show this analytically, let's consider a network after learning $T$ sequential tasks with weights of the network at any layer, $l$ expressed as :

$$\boldsymbol{W}_T^l = \boldsymbol{W}_1^l + \sum_{i=1}^{T-1} \Delta \boldsymbol{W}_{i \to i+1}^l = \boldsymbol{W}_1^l + \Delta \boldsymbol{W}_{1 \to T}^l. \tag{11}$$

Here, $\boldsymbol{W}_1^l$ is the weights after task 1 and $\Delta \boldsymbol{W}_{1 \to T}^l$ is the change of weights from task 1 to T. Weight update with our method ensures that $\Delta \boldsymbol{W}_{1 \to T}^l$ lie in the orthogonal space of the data (representations) of task 1. Linear operation at layer $l$ with data from task 1 ($\boldsymbol{x}_1$) would produce: $\boldsymbol{W}_T^l \boldsymbol{x}_1^l = \boldsymbol{W}_1^l \boldsymbol{x}_1^l + \Delta \boldsymbol{W}_{1 \to T}^l \boldsymbol{x}_1^l$. If $\Delta \boldsymbol{W}_{1 \to T}^l \boldsymbol{x}_1^l = 0$, then the output of the network for task 1 data after learning task $T$ will be the same as the output after learning task 1 (*i.e.* $\boldsymbol{W}_T^l \boldsymbol{x}_1^l = \boldsymbol{W}_1^l \boldsymbol{x}_1^l$), that means no interference for task 1. We define $\Delta \boldsymbol{W}_{1 \to T}^l \boldsymbol{x}_1^l$ as the **interference activation** for task 1 at layer $l$ (for any task, $\tau < T$: $\Delta \boldsymbol{W}_{\tau \to T}^l \boldsymbol{x}_\tau^l$). As discussed above, degree of such interference is dictated by $\epsilon_{th}$. Figure 3(a)-(b) (and Figure 5 in appendix) show histograms (distributions) of interference activations at each layer of the network for split CIFAR-100 experiment. For lower value of $\epsilon_{th}$, these distributions have higher variance (spread) implying high interference, whereas with increasing value of $\epsilon_{th}$, the variance reduces around the (zero) mean value. As a direct consequence, as shown in Figure 3(c), backward transfer reduces for increasing $\epsilon_{th}$ with improvement in accuracy.

## 8 CONCLUSION

In this paper we propose a novel continual learning algorithm that finds important gradient subspaces for the past tasks and minimizes catastrophic forgetting by taking gradient steps orthogonal to these subspaces when learning a new task. We show how to analyse the network representations to obtain minimum number of bases of these subspaces by which past information is preserved and learnability for the new tasks is ensured. Evaluation on diverse image classification tasks with different network architectures and comparisons with state-of-the-art algorithms show the effectiveness of our approach in achieving high classification performance while mitigating forgetting. We also show our algorithm is fast, makes efficient use of memory and is capable of learning long sequence of tasks in deeper networks preserving data privacy.

ACKNOWLEDGMENTS

This work was supported in part by the National Science Foundation, Vannevar Bush Faculty Fellowship, Army Research Office, MURI, and by Center for Brain Inspired Computing (C-BRIC), one of six centers in JUMP, a Semiconductor Research Corporation program sponsored by DARPA.

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

## A    APPENDIX

## B    ALGORITHM

### B.1    INPUT AND GRADIENT SPACES (CONT.)

Since, the batch loss is the summation of the losses due to individual examples, the total batch loss for $n$ samples can be expressed as

$$L_{batch} = \sum_{i=1}^{n} L_i = \sum_{i=1}^{n} \frac{1}{2} ||\boldsymbol{W}\boldsymbol{x}_i - \boldsymbol{y}_i||_2^2. \tag{12}$$

The gradient of this loss with respect to weights can be expressed as

$$\nabla_{\boldsymbol{W}} L_{batch} = \boldsymbol{\delta_1}\boldsymbol{x}_1^T + \boldsymbol{\delta_2}\boldsymbol{x}_2^T + ... + \boldsymbol{\delta_n}\boldsymbol{x}_n^T. \tag{13}$$

The gradient update will remain in the subspace spanned by the $n$ input examples.

### B.2    ALGORITHM PSEUDO CODE

---
**Algorithm 1** Algorithm for Continual Learning with GPM
---
1: **function** TRAIN ($f_{\boldsymbol{W}}$, $\mathcal{D}^{train}$, $\alpha$, $\epsilon_{th}$ )
2:   Initialize, $\boldsymbol{M}^l \leftarrow$ [ ], for all $l = 1, 2, ....L$   // till L-1 if multi-head setting
3:   $\mathcal{M} \leftarrow \{(\boldsymbol{M}^l)_{l=1}^L\}$
4:   $\boldsymbol{W} \leftarrow \boldsymbol{W}_0$
5:   **for** $\tau \in 1, 2, ....., T$ **do**
6:     **repeat**
7:       $B_n \sim \mathcal{D}_\tau^{train}$   // sample a mini-batch of size $n$ from task $\tau$
8:       gradient, $\nabla_{\boldsymbol{W}} L_\tau \leftarrow$ SGD($B_n, f_{\boldsymbol{W}}$)
9:       $\nabla_{\boldsymbol{W}} L_\tau \leftarrow$ PROJECT($\nabla_{\boldsymbol{W}} L_\tau, \mathcal{M}$)   // see equation (6, 7)
10:      $\boldsymbol{W} \leftarrow \boldsymbol{W} - \alpha\nabla_{\boldsymbol{W}} L_\tau$
11:    **until** convergence
12:
13:    // Update Memory (GPM)
14:    $B_{n_s} \sim \mathcal{D}_\tau^{train}$   // sample a mini-batch of size $n_s$ from task $\tau$
15:    // construct representation matrices for each layer by forward pass (section 5)
16:    $\mathcal{R}_\tau \leftarrow$ forward($B_{n_s}, f_{\boldsymbol{W}}$), where $\mathcal{R}_\tau = \{(\boldsymbol{R}_\tau^l)_{l=1}^L\}$
17:    **for** layer, $l = 1, 2, ...L$ **do**
18:      $\hat{\boldsymbol{R}}_\tau^l \leftarrow$ PROJECT($\boldsymbol{R}_\tau^l, \boldsymbol{M}^l$)   // see equation (8)
19:      $\hat{\boldsymbol{U}}_\tau^l \leftarrow$ SVD($\hat{\boldsymbol{R}}_\tau^l$)
20:      $k \leftarrow$ criteria($\hat{\boldsymbol{R}}_\tau^l, \boldsymbol{R}_\tau^l, \epsilon_{th}^l$)   // see equation (9)
21:      $\boldsymbol{M}^l \leftarrow [\boldsymbol{M}^l, \hat{\boldsymbol{U}}_\tau^l[0:k]]$
22:    **end for**
23:  **end for**
24:  **return** $f_{\boldsymbol{W}}, \mathcal{M}$
25: **end function**
---

## C    EXPERIMENTAL DETAILS

### C.1    DATASET STATISTICS

Table 4 and Table 5 show the summary of the datasets used in the experiments.

Table 4: Dataset Statistics.

|  | PMNIST | Split CIFAR-100 | Split-miniImageNet |
|---|---|---|---|
| num. of tasks | 10 | 10 | 20 |
| input size | $1 \times 28 \times 28$ | $3 \times 32 \times 32$ | $3 \times 84 \times 84$ |
| # Classes/task | 10 | 10 | 5 |
| # Training samples/tasks | 54,000 | 4,750 | 2,375 |
| # Validation Samples/tasks | 6,000 | 250 | 125 |
| # Test samples/tasks | 10,000 | 1,000 | 500 |

Table 5: 5-Datasets Statistics. For the datasets with monochromatic images, we replicate the image across all RGB channels so that size of each image becomes $3 \times 32 \times 32$.

|  | CIFAR-10 | MNIST | SVHN | Fashion MNIST | notMNIST |
|---|---|---|---|---|---|
| # Classes | 10 | 10 | 10 | 10 | 10 |
| # Training samples | 47,500 | 57,000 | 69,595 | 57,000 | 16,011 |
| # Validation Samples | 2,500 | 3,000 | 3,662 | 3,000 | 842 |
| # Test samples | 10,000 | 10,000 | 26,032 | 10,000 | 1,873 |

## C.2    ARCHITECTURE DETAILS

**AlexNet-like architecture:** This is the same architecture used by Serrà et al. (2018) with batch normalization added in each layer except the classifier layer. The network consists of 3 convolutional layers of 64, 128, and 256 filters with $4 \times 4$, $3 \times 3$, and $2 \times 2$ kernel sizes, respectively, plus two fully connected layers of 2048 units each. Rectified linear units is used as activations, and $2 \times 2$ max-pooling after the convolutional layers. Dropout of 0.2 is used for the first two layers and 0.5 for the rest.

**Reduced ResNet18 architecture:** This is the similar architecture used by Lopez-Paz & Ranzato (2017). For miniImageNet experiment, we use convolution with stride 2 in the first layer. For both miniImageNet and 5-Datasets experiments we replace the $4 \times 4$ average-pooling before classifier layer with $2 \times 2$ average-pooling.

All the networks use ReLU in the hidden units and softmax with cross entropy loss in the final layer.

## C.3    CIFAR-100 SUPERCLASS EXPERIMENT

For this experiment, similar to APD (Yoon et al., 2020), we use a modified LeNet-5 architecture with 20-50-800-500 neurons. All the baseline results are reported from APD. Like APD, we do not use any data augmentation or preprocessing. We keep $5\%$ of training data from each task for validation. We train the network with our algorithm with batch size of 64 and initial learning rate of 0.01. We Train each task for a maximum of 50 epochs with decay schedule and early termination strategy similar to Serrà et al. (2018). We use $\epsilon_{th} = 0.98$ for all the layers and increasing the value of $\epsilon_{th}$ by 0.001 for each new tasks.

## C.4    BASELINE IMPLEMENTATIONS

GEM (Lopez-Paz & Ranzato, 2017), A-GEM (Chaudhry et al., 2019a), ER_Res (Chaudhry et al., 2019b) and OWM (Zeng et al., 2018) are implemented from their respective official implementations. EWC and HAT are implemented from the official implementation provided by Serrà et al. (2018). While, OGD is implemented from adapting the code provided by Bennani et al. (2020).

## C.5 Threshold hyperparameter

As discussed in section 5 and section 7, the threshold hyperparameter, $\epsilon_{th}$ controls the degree of interference through the approximation of space of significant representations of the past tasks. Since in neural network, characteristics of learned representations vary for different architectures and different dataset, using the same value of $\epsilon_{th}$ may not be useful in capturing the similar space of significance. In our experiments we use $\epsilon_{th}$ in the range of 0.95 to 1. For PMNIST experiment, as discussed in section 7, we use $\epsilon_{th} = 0.95$ in the first layer and 0.99 in the other layers. For split CIFAR-100 experiment, we use $\epsilon_{th} = 0.97$ for all the layers and increasing the value of $\epsilon_{th}$ by 0.003 for each new tasks. For split miniImageNet experiment, we use $\epsilon_{th} = 0.985$ for all the layers and increasing the value of $\epsilon_{th}$ by 0.0003 for each new tasks. For experiment with 5-Datasets, we use $\epsilon_{th} = 0.965$ for all the layers across all the tasks.

## C.6 Training Time Measurement

We measured per epoch training times (in Figure 2(b)) for computation in NVIDIA GeForce GTX 1060 GPU. For ten sequential tasks in PMNIST experiment, we computed per epoch training time for each task and reported the average value over all the tasks.

Training time for different algorithms reported in Table 2(a) for PMNIST tasks were measured on a Single NVIDIA GeForce GTX 1060 GPU. For all the other datasets, training time for different algorithms reported in Table 2(b) were measured on a Single NVIDIA GeForce GTX 1080 Ti GPU.

## C.7 List of hyperparameters

Table 6: List of hyperparameters for the baselines and our approach. Here, 'lr' represents (initial) learning rate. In the table we represent PMNIST as 'perm', 10-Split CIFAR-100 as 'cifar', Split miniImageNet as 'minImg' and 5-Datasets as '5data'.

| Methods | Hyperparameters |
|---------|-----------------|
| OGD | lr : 0.001 (perm)
# stored gradients : 200/task (perm) |
| OWM | lr : 0.01 (cifar), 0.3 (perm) |
| GEM | lr : 0.1 (perm)
memory size (samples) : 1000 (perm)
memory strength, $\gamma$ : 0.5 (perm) |
| A-GEM | lr : 0.05 (cifar), 0.1 (perm, minImg, 5data)
memory size (samples) : 1000 (perm), 2000 (cifar), 500 (minImg), 3000 (5data) |
| ER_Res | lr : 0.05 (cifar), 0.1 (perm, minImg, 5data)
memory size (samples) : 1000 (perm), 2000 (cifar), 500 (minImg), 3000 (5data) |
| EWC | lr : 0.03 (perm, minImg, 5data), 0.05 (cifar)
regularization coefficient : 1000 (perm), 5000 (cifar, minImg, 5data) |
| HAT | lr : 0.03 (minImg), 0.05 (cifar), 0.1 (5data)
$s_{max}$ : 400 (cifar, minImg, 5data)
$c$ : 0.75 (cifar, minImg, 5data) |
| Multitask | lr : 0.05 (cifar), 0.1 (perm, minImg, 5data) |
| GPM (ours) | lr : 0.01 (perm, cifar), 0.1 (minImg, 5data)
$n_s$ : 100 (minImg, 5data), 125 (cifar), 300 (perm) |

## C.8   GPM SIZE

As discussed in section 4, the gradient update will lie in the span of input vectors ($x$) in fully connected layers, whereas the gradient updates of the convolutional filters will lie in the space spanned by the input patch vectors ($p$). Therefore, each basis stored in the GPM for a particular layer, $l$ will have the same dimension as $x^l$ or $p^l$. Thus, for any particular layer, GPM matrix, $M^l$ can have a maximum size of : size($x^l$) × size($x^l$) or size($p^l$) × size($p^l$). Maximum size of the GPM, which we refer as GPM_Max, is computed by including GPM matrices ($M^l$) from all the layers. Thus the size of GPM_Max is fixed by the choice of network architecture. In Table 7, we show the maximum size of GPM matrix ($M^l$) for each layers for the architectures that we have used in our experiments along with the size of GPM_Max.

Table 7: Size of GPM matrices for each layer for the architectures used in our experiments. Maximum sizes of the GPM in terms of number of parameters are also given.

| Network | Size of maximum $M^l$ | GPM_Max (parameters) |
|---|---|---|
| MLP (3 layers) | $784 \times 784$, $100 \times 100$, $100 \times 100$ | 0.63M |
| AlexNet (5 layers) | $48 \times 48$, $576 \times 576$, $512 \times 512$, $1024 \times 1024$, $2048 \times 2048$ | 5.84M |
| ResNet18 (17 layers+ 3 short-cut connections) | $27 \times 27$, $180 \times 180$, $180 \times 180$, $180 \times 180$, $180 \times 180$, $180 \times 180$, $360 \times 360$, $20 \times 20$, $360 \times 360$, $360 \times 360$, $360 \times 360$, $720 \times 720$, $40 \times 40$, $720 \times 720$, $720 \times 720$, $720 \times 720$, $1440 \times 1440$, $80 \times 80$, $1440 \times 1440$, $1440 \times 1440$ | 8.98M |

## D   ADDITIONAL RESULTS

### D.1   RESULT TABLES

Table 8 contains the additional results for PMNIST experiment in single-epoch setting along with the standard deviation values for the results shown in Table 1(a) for multi-epoch (5 epoch) setting. Method that does not adhere to CL setup is indicated by (*) in the table. Results are reported from 5 different runs.

Table 8: Continual learning on PMNIST in single-epoch and multi-epoch setting.

| Methods | 1 Epoch | | 5 Epochs | |
|---|---|---|---|---|
| | ACC (%) | BWT | ACC (%) | BWT |
| OGD | $85.18 \pm 0.29$ | $-0.06 \pm 0.00$ | $82.56 \pm 0.66$ | $-0.14 \pm 0.01$ |
| OWM | $90.55 \pm 0.15$ | $-0.01 \pm 0.00$ | $90.71 \pm 0.11$ | $-0.01 \pm 0.00$ |
| GEM | $88.65 \pm 0.27$ | $-0.07 \pm 0.00$ | $83.38 \pm 0.56$ | $-0.15 \pm 0.01$ |
| A-GEM | $87.80 \pm 0.16$ | $-0.08 \pm 0.00$ | $83.56 \pm 0.16$ | $-0.14 \pm 0.00$ |
| ER_Res | $90.63 \pm 0.27$ | $-0.05 \pm 0.00$ | $87.24 \pm 0.53$ | $-0.11 \pm 0.01$ |
| EWC | $88.27 \pm 0.39$ | $-0.04 \pm 0.01$ | $89.97 \pm 0.57$ | $-0.04 \pm 0.01$ |
| GPM (ours) | $91.74 \pm 0.15$ | $-0.03 \pm 0.00$ | $93.91 \pm 0.16$ | $-0.03 \pm 0.00$ |
| Multitask* | $95.21 \pm 0.01$ | - | $96.70 \pm 0.02$ | - |

Table 9: Continual learning on different datasets along with the standard deviation values for the results shown in Table 1(b).

| Methods | CIFAR-100 | | miniImageNet | | 5-Datasets | |
|---|---|---|---|---|---|---|
| | ACC (%) | BWT | ACC (%) | BWT | ACC (%) | BWT |
| OWM | $50.94 \pm 0.60$ | - $0.30 \pm 0.01$ | - | - | - | - |
| EWC | $68.80 \pm 0.88$ | - $0.02 \pm 0.01$ | $52.01 \pm 2.53$ | - $0.12 \pm 0.03$ | $88.64 \pm 0.26$ | - $0.04 \pm 0.01$ |
| HAT | $72.06 \pm 0.50$ | - $0.00 \pm 0.00$ | $59.78 \pm 0.57$ | - $0.03 \pm 0.00$ | $91.32 \pm 0.18$ | - $0.01 \pm 0.00$ |
| A-GEM | $63.98 \pm 1.22$ | - $0.15 \pm 0.02$ | $57.24 \pm 0.72$ | - $0.12 \pm 0.01$ | $84.04 \pm 0.33$ | -$0.12 \pm 0.01$ |
| ER_Res | $71.73 \pm 0.63$ | - $0.06 \pm 0.01$ | $58.94 \pm 0.85$ | - $0.07 \pm 0.01$ | $88.31 \pm 0.22$ | - $0.04 \pm 0.00$ |
| GPM (ours) | $72.48 \pm 0.40$ | - $0.00 \pm 0.00$ | $60.41 \pm 0.61$ | - $0.00 \pm 0.00$ | $91.22 \pm 0.20$ | - $0.01 \pm 0.00$ |
| Multitask* | $79.58 \pm 0.54$ | - | $69.46 \pm 0.62$ | - | $91.54 \pm 0.28$ | - |

## D.2 $k$ VALUES

Table 10 (a) and (b) show the number of new bases added at each layer per PMNIST and 10-split CIFAR-100 task respectively. Total number of bases in the GPM after learning all the tasks is also given.

Table 10: Number of new bases ($k$) added to the GPM at different layers after each (a) PMNIST task and (b) 10-split CIFAR-100 task (for a random seed configuration).

| (a) | | | | (b) | | | | | |
|---|---|---|---|---|---|---|---|---|---|
| | $k$ | | | | | $k$ | | | |
| Task ID | FC1 | FC2 | FC3 | Task ID | $\epsilon_{th}$ | Conv1 | Conv2 | Conv3 | FC1 | FC2 |
| 1 | 81 | 60 | 41 | 1 | 0.970 | 7 | 125 | 197 | 80 | 98 |
| 2 | 71 | 22 | 19 | 2 | 0.973 | 3 | 44 | 74 | 81 | 101 |
| 3 | 70 | 11 | 11 | 3 | 0.976 | 0 | 20 | 29 | 71 | 93 |
| 4 | 61 | 4 | 7 | 4 | 0.979 | 1 | 21 | 26 | 76 | 99 |
| 5 | 57 | 2 | 4 | 5 | 0.982 | 2 | 33 | 28 | 73 | 98 |
| 6 | 50 | 1 | 2 | 6 | 0.985 | 0 | 16 | 19 | 71 | 98 |
| 7 | 46 | 0 | 2 | 7 | 0.988 | 0 | 16 | 14 | 71 | 99 |
| 8 | 41 | 0 | 1 | 8 | 0.991 | 2 | 41 | 26 | 76 | 104 |
| 9 | 35 | 0 | 1 | 9 | 0.994 | 6 | 56 | 34 | 84 | 109 |
| 10 | 31 | 0 | 0 | 10 | 0.997 | 1 | 45 | 26 | 86 | 113 |
| Total | 543 | 100 | 88 | Total | | 22 | 417 | 473 | 769 | 1012 |

Table 11: Continual learning of Digit dataset tasks in class-incremental learning setup (Kamra et al., 2017). (†) denotes the result reported from DGDMN.

| | Methods | | | |
|---|---|---|---|---|
| Metric | EWC† | DGR† | DGDMN† | GPM (ours) |
| ACC (%) | 10.00 | 59.60 | **81.80** | 70.67 |
| BWT | - 1.00 | - 0.43 | **- 0.15** | - 0.26 |

## D.3 CLASS-INCREMENTAL LEARNING

In this section, we evaluate our algorithm in a class-incremental learning setup (Rebuffi et al., 2017), where disjoint classes are learned one by one and classification is performed within all the learned classes without task hint (Kamra et al., 2017). This setup is different (Hsu et al., 2018) from the (single-head/multi-head) evaluation setups used throughout this paper. Also, this scenario is very challenging and often infeasible for the regularization (HAT, EWC etc.) and expansion-based (DEN, APD etc.) methods which do not use old data replay. Using the experimental setting similar to DGDMN (Kamra et al., 2017), we implemented the 'Digit dataset' experiment where a single class

of MNIST digit is learned per task. Results are listed in Table 11, where the baselines are reported from DGDMN. While EWC forgets catastrophically, we perform better than DGR (Shin et al., 2017), which employs data replay through old data generation. DGDMN, an improved data replay method, outperforms all. In this setup, we believe a subset of old data replay either from storage or via generation is inevitable for attaining better performance with minimal forgetting (Rebuffi et al., 2017; Rajasegaran et al., 2019).

## D.4 ADDITIONAL PLOTS

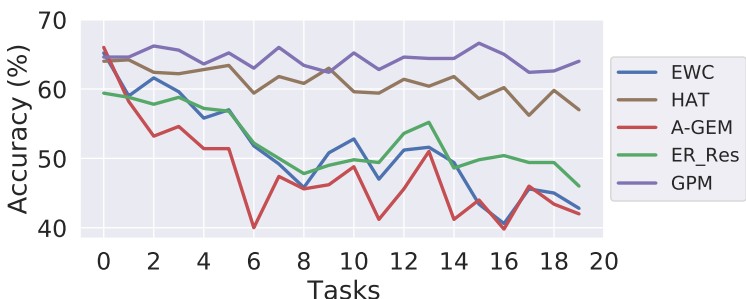

Figure 4: Evolution of task 1 accuracy over the course of incremental learning of 20 sequential tasks from miniImageNet dataset. Learned accuracy in our method remains stable throughout learning.

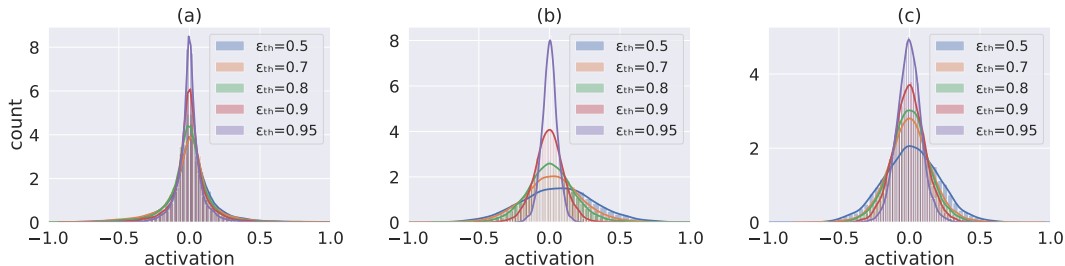

Figure 5: Illustration of how threshold hyperparameter controls the degree of interference at (a) Conv layer 1 (b) Conv layer 3 (c) FC layer 1 with the histogram plots of interference activations from Split CIFAR-100 experiment. With increasing $\epsilon_{th}$, spread of the inference activation decreases resulting in minimization of forgetting.

