# OpenReview forum: "Gradient Projection Memory for Continual Learning"
_ICLR.cc/2021/Conference — ICLR 2021 Oral_

### Official Review · AnonReviewer2 · 2020-10-21
**Great method and paper applied for one of the most basic continual learning settings**

**Rating:** 8
**Confidence:** 5

**Review:**

Summary:
The paper proposes one of the most scalable approaches to sequential continual learning with known task boundaries and related tasks, while taking steps towards enforcing data privacy and removing some of the task label constraints. At all levels in expressive deep models, SVD is used on learned representations to identify important bases of task gradient spaces and a memory is populated with such directions. Learning progresses only in directions orthogonal to gradient memory. Several recent evaluation methodologies are used to empirically validate the approach with significant success.


Strong points:
- Principled and relatively simple approach, yet scalable to interesting deep models.
- Manageable computational overhead.
- Memory of gradients introduces a layer of data privacy, and advantage compared to many other memory-based approaches.
- Strong empirical results, although not apples-to-apples in all cases, see the comment posted earlier.
- Clearly written paper.
- Analysis of scalability in terms of memory and computation is a welcome bonus.


Weak points:
- The sequential task learning setup is of limited practical use and not too realistic.
- It’s hard to say how the algorithm would be used in practical continual learning settings, e.g. reinforcement learning of single tasks without forgetting, or where clear task boundaries do not exist.
- Little effort is made to see the approach relevance for non-similar sequential task learning, where there could be interference between learned representations in terms of negative forward transfer.


Recommendation and Rationale:

I strongly recommend acceptance because the method is simple, practical and the paper is well written both in terms of clarity and also analysis.


Further questions:
- Do you see any positive forward transfer, e.g. later tasks are learned quicker due to previous tasks?
- What are the limitations and roadblocks to extending this method to sequences of hundreds of tasks which are not necessarily related? Please discuss in the manuscript.

---

> ### Author Response · Authors · 2020-11-19
> **Author's Response to Reviewer 2 (2/2)**
>
> ##### **Clarification on multitask baseline**
> **Response:** In Multitask baseline, all tasks are jointly learned (in a multitask learning fashion) using the entire dataset at once in a single network. HAT, A-GEM and ER-Res used this baseline to show the upper-bound on the average accuracy on all tasks. In APD, STL baseline is used to show such upper bound. STL is different from Multitask because in STL each task is independently learned in a separate network. Since, in our paper, for 10-split CIFAR-100 experiment we used a bigger architecture, AlexNet as opposed to APD’s LeNet-5 and trained with a different learning rate schedule, our average accuracy upper bound (Multitask) over 10-tasks is significantly higher than APD’s (STL). We have clarified the definition of multitask baseline in the revised manuscript.
> ##### **Discussion on Results**
> **Response:** For training our method and the baselines we did not use any data augmentation. Thus, compared to APD’s 10-split CIFAR-100 experiments, the accuracy improvements in our CIFAR-100 experiments (for both our methods and baselines) come from training better network architecture (AlexNet) with higher learning rate schedules and optimum hyperparameters (table 6 in GPM). Moreover, with direct comparison with APD (in LeNet-5) we have shown that our algorithm induces more sharing and yields better performance than APD through efficient use of the GPM.
>
> [1] Scalable and Order-robust Continual Learning with Additive Parameter Decomposition. Jaehong Yoon, Saehoon Kim, Eunho Yang, Sung Ju Hwang.\
> [2] Continual Learning with Adaptive Weights (CLAW). Tameem Adel, Han Zhao, Richard E. Turner.\
> [3] An Adaptive Random Path Selection Approach for Incremental Learning. Jathushan Rajasegaran, Munawar Hayat, Salman Khan, Fahad Shahbaz Khan, Ling Shao, Ming-Hsuan Yang.

---

> ### Author Response · Authors · 2020-11-19
> **Author's Response to Reviewer 2 (1/2)**
>
> We thank the reviewer for taking the time and reviewing our paper. We are glad to hear his/her positive comments and appreciation for our work. We provide answers to the reviewer’s questions below. Also, we have updated the manuscript according to the suggestions from the reviewers. Changes are marked in blue.
>
> **Q1:** *"Do you see any positive forward transfer, e.g. later tasks are learned quicker due to previous tasks?"* \
> **Answer:** We do observe a trend (e.g. in CIFAR-100 experiment) that later tasks obtain higher classification accuracy with fewer epochs than the earlier tasks.
>
> **Q2:**  *"What are the limitations and roadblocks to extending this method to sequences of hundreds of tasks which are not necessarily related? "* \
> **Answer:** Our algorithm mitigates catastrophic forgetting in continual learning within a fixed capacity network. The GPM size is also determined by the chosen network architecture. With each new task, new bases are added to the GPM, which in turns increases the volume of gradient spaces along which gradient steps are restricted. With very long sequence of tasks GPM size is expected to reach at its maximum which would mean no new learning will be possible for the incoming tasks. The rate at which GPM gets populated with new bases depends on the task similarity. Similar task is expected to share similar representation across layers; thus our algorithm would add smaller number of new bases to GPM. In contrast, if the tasks are less or non-similar, GPM will get populated faster.  Therefore, the ability of learning sequences of hundreds of tasks (which are not necessarily related) with our method will be limited by the chosen network capacity. We have added a summary of this discussion in the revised manuscript (section 7).
> ---
> We provide the clarification on SOTA and experimental setup from reviewer’s previous comments below:
> ##### **On direct comparison with SOTA baselines such as APD[1], CLAW[2], Adaptive RPS-Net[3]**
>  **Response:** In contrast to our method, CLAW uses a probabilistic model and variational inference framework for continual learning. Whereas, Adaptive RPS-Net learns continually through a hybrid scheme involving episodic memory, regularization, and network architecture expansion. However, Adaptive RPS-Net uses a different (class incremental learning) evaluation protocol which prevents direct comparison with our method.  On the other hand, APD has shown to achieve SOTA performance with a moderate increase in network capacity. Moreover, it uses a similar experimental/evaluation setup to our method, thus warrants a direct comparison. To show a fair comparison with the APD, we implemented our method in the network (LeNet-5) used in APD. The results of CIFAR-100 experiments are given below:
>
> **Experiment 1: CIFAR-100 Split (10 tasks)** \
> *Methods \ Metrics | Capacity (%) | Accuracy (%)|* \
> STL   | 1000% | 63.75% | \
> APD  | 135%   | 60.74% | \
> GPM | 100%   | 61.81% |
>
> **Experiment 2: CIFAR-100 Superclass (20 tasks)** \
> *Methods \ Metrics | Capacity (%) | Accuracy (%)|* \
> STL   | 2000% | 61.00% | \
> APD  | 130%   | 56.81% | \
> GPM | 100%   | 57.72% |
>
> We did not use dropout or weight decay. Like, APD we did not use any data augmentation or preprocessing. Results for STL and APD are taken from the APD paper (table 1). For both datasets, reported results (of APD and GPM) are the average accuracy of 5 recommended task sequences given in APD. In both CIFAR100-100 split (10 tasks) and CIFAR-100 Superclass (20 tasks) experiments, our method outperforms APD using 35% and 30% fewer parameters respectively with less than 1% forgetting. These results show GPM induces more sharing between tasks than APD.
>
> We thank the reviewer for suggesting the CIFAR-100 superclass experiment. We have added this experiment and comparison (Table 3) with the expansion-based CL baselines (including APD) in the revised version of the paper. We have also added experimental details in appendix C.3.

---

### Official Review · AnonReviewer1 · 2020-10-27
**Interesting approach but kind of incremental. Experiments are good, but it lacks running time comparison.**

**Rating:** 6
**Confidence:** 5

**Review:**

This manuscript proposes a new approach for continual learning problems. The key idea is to maintain bases of subspaces using SVD in the Gradient Projection Memory (GPM), in which the update direction is orthogonal to the gradient subspaces deemed important for the past tasks. Image classification experiment was conducted to justify its better empirical performance in practice. Overall, the paper is well-written. I have the following comments.

1. The contribution of this manuscript is kind of incremental compared with OGD (Farajtabar et al. 2020). From my understanding, the main improvement is using SVD to store the bases, which basically trades computational efficiency for memory. In addition, it is claimed that OGD only works for small learning rate. Why using SVD helps using a larger learning rate? It is not explained in the paper.
2.  Is it possible to report running time of the proposed approach and compare with other approaches? SVD is very expensive in the high-dimensional setting such as deep neural networks. It might be impractical due to running time concerns.
3. What is the value of $k$ in the experiments (in inequality (9)) for different $\epsilon_{th}^l$? It is better to report these values. In addition, when $k$ becomes large, the update rule (6) and (7) may become computationally expensive.
4. It is mentioned that “our proposed approach can perform inference without the task identity of test samples”, but with no explanations. It is desired to illustrate the meaning in the main paper.
5. The literature review is not sufficient. There are several recent papers which also change the update direction using gradient projection. It is desired to see the difference between the proposed method and other relevant approaches. For example,

	[1]. Yu et al. Gradient surgery for multi-task learning. NeurIPS 2020.

	[2]. Guo et al. Improved schemes for episodic memory-based lifelong learning. NeurIPS 2020.

========POST REBUTTAL=========

I would like to thank the authors to answer my questions. It addressed most of my concerns. Hence I increase my score to 6.

---

> ### Author Response · Authors · 2020-11-18
> **Author's Response to Reviewer 1 (2/2)**
>
> **Reviewer’s Comment 3:** *" What is the value of k in the experiments (in inequality (9)) for different ϵthl? ... In addition, when k becomes large, the update rule (6) and (7) may become computationally expensive.".* \
> **Response:** Our reported GPM size (in Figure. 2) implicitly captures the cumulative value of $k$ after all the tasks. Additionally, following reviewer’s suggestion, we have added the values of $k$ for the PMNIST and 10-split CIFAR-100 experiments in Table 10 in the appendix of the revised manuscript. \
> In our implementation, we precomputed the projection matrices ($MM^T$) (in equation (6,7)) from GPM bases at the start of each task and reused the values during training. Moreover, these projection matrices are of fixed and reasonable dimensions (see appendix section C.8) for efficient computation in GPUs. Therefore, we do not see significant time differences with increasing value of $k$ during training with update rule (6,7).
>
> **Reviewer’s Comment 4:** *" It is mentioned that “our proposed approach can perform inference without the task identity of test samples”, but with no explanations".* \
> **Response:** We thank the reviewer for pointing this out. In our PMNIST experiment, we perform task-hint free inference, whereas in all the other experimental settings (following the setup in baselines) we use task hint to select classifier head during inference. We made this statement to highlight the findings from the PMNIST experiment. However, we acknowledge that our statement gives a general feeling that in all the experiments we are using task-free inference, which is not the case. Hence, we remove these statements in our revised manuscript. We show in the PMNIST experiment section how our method performs task-hint free inference in that experimental/evaluation setup.
>
> **Reviewer’s Comment 5:** *“The literature review is not sufficient. Show the difference between the Proposed method and [1],[2].”*\
> **Response:** Yu et al. (2020) [1] use gradient projection in ‘multitask learning’ setting where data from all the tasks are available for necessary reference gradient generation during (joint) training.  In contrast, we operate in ‘continual learning’ setting where data from only one task is available at a time (and GPM provides necessary direction for gradient projection).
> Guo et al. [2] proposes a unified view of episodic memory-based CL methods, including GEM and A-GEM and improves performance over these methods utilizing loss-balancing update rule. In contrast to this method, we do not store raw data in the memory thus provide data privacy. We have added this reference to the related works section in our revised manuscript.
>
> [1] Yu et al. Gradient surgery for multi-task learning. NeurIPS 2020.\
> [2] Guo et al. Improved schemes for episodic memory-based lifelong learning. NeurIPS 2020.

---

> ### Author Response · Authors · 2020-11-18
> **Author's Response to Reviewer 1 (1/2)**
>
> We’d like to thank the reviewer for taking the time to read our submission and for his/her comments.  We address the reviewer’s comments below. Also, we have updated the manuscript according to the suggestions from the reviewers. Changes are marked in blue.
>
> **Reviewer’s comment 1(a):** *"The contribution of this manuscript is kind of incremental compared with OGD (Farajtabar et al. 2020). From my understanding, the main improvement is using SVD to store the bases, which basically trades computational efficiency for memory."* \
> **Response:** We would like to take this opportunity to clarify and highlight the difference between our method and OGD.  To minimize forgetting, both OGD and GPM take gradient steps in the orthogonal direction to the gradient space important for the past tasks. However, the formulation of such gradient spaces in OGD and GPM are different. Most importantly, the dimensionality of the bases of these respective gradient spaces in OGD and GPM are different hence these bases cannot be trivially related to one another by SVD.
> - In OGD, after learning each task, ‘gradient directions’ are computed from the network predictions on the samples of the current task and stored in the memory. To minimize forgetting, new tasks are learned by moving in the orthogonal directions to space spanned by these ‘gradient directions’.
> - In contrast, in GPM, after learning each task, we collect the ‘input representations’ from the sample of the current task (section 5). We apply SVD on these ‘input representations’ and find the bases of representation. We show how input and gradient spaces are related (section 4), which allows us to use these bases of representations to describe a subspace in the gradient space. These bases are stored as GPM and new tasks are learned in the orthogonal directions of the space described by GPM.
> - For example, let’s consider a FC layer of a network with weight W ($m\times n$), where $m$ and $n$ are the number of output and input nodes respectively. Total parameters in the layer is, $p=m*n$. In OGD, each ‘gradient direction’ will be $p$ dimensional.  Thus, for $n_s$ input samples, $n_s$ such $p$ dimensional gradient vectors will span the gradient space. Whereas, in GPM, for $n$ dimensional inputs, bases of ‘input representation’ hence the bases of gradient space will be $n$ dimensional. Thus, in GPM we not only describe different subspace than OGD but also bases in GPM formulation are significantly lower-dimensional which enables computationally efficient projection and (GPM) memory management.
>
> **Reviewer’s comment 1(b):** *"In addition, it is claimed that OGD only works for small learning rate. Why using SVD helps using a larger learning rate? It is not explained in the paper."* \
> **Response:** Since OGD uses gradient directions to describe important gradient spaces for the past tasks, gradient updates need to obey locality assumptions (*Farajtabar et al. 2020*), which constrains OGD to use small learning rates. In contrast to OGD, we do not use gradient directions to describe our important gradient spaces, thus we do not need to adhere to such locality assumption. For a suitable threshold hyperparameter (section 7), our method changes weights in the orthogonal direction of the past significant representations preserving old input weight correlations. Our experiments show that we can use a broad range of learning rate hence scale our algorithm to SOTA architectures.  We have added a brief explanation about the learning rate in the revised manuscript.
>
> **Reviewer’s comment 2:**  *"Is it possible to report running time of the proposed approach and compare with other approaches? SVD is very expensive in the high-dimensional setting such as deep neural networks. It might be impractical due to running time concerns."* \
> **Response:** We thank the reviewer for this suggestion. In Table 2 of the revised manuscript, we report the training time of our algorithm and compare it with the baselines. The necessary discussions are also added. Our algorithm is fast with little to manageable overhead due to GPM update using SVD. Several factors contribute to fast runtime. Our formulation uses GPM bases and projection matrices ($MM^T$) of reasonable dimensions (see appendix section C.8). Precomputation of these bases and the projection matrices at the start of each task leads to faster per-epoch training in GPU. This gain in time essentially compensates for the extra time required for GPM update through SVD.  For example, in 5-dataset experiment, where ResNet18 is used, our algorithm runs faster than others using only 6% of the training time in GPM update. Therefore, our method despite relying on SVD computation is fast and scalable for deeper networks.

---

### Official Review · AnonReviewer4 · 2020-10-28
**New method for continual learning, but needs more thorough evaluation**

**Rating:** 8
**Confidence:** 5

**Review:**

Summary:

This work targets learning multi-class classifiers in the continual learning setting. The key idea is to learn new tasks by taking gradient steps in directions orthogonal to the gradient subspaces marked as crucial for previous past tasks. The method employs SVD after learning each task to find the crucial subspaces (which it calls as Core Gradient Subspaces) and stores them in a memory. Reasonable quantitative and qualitative evaluation has been performed to compare the method against existing SOTA baselines.


Review:

1. The paper is overall clearly written and the method is adequately described.

2. The work is relevant to audience at ICLR and provides reasonable details for reproducibility.

3. Relationship to previous works has been explained and relevant literature has been cited appropriately.


Strengths:

1. GPM shows very little degradation in performance on past tasks and is very resilient to forgetting.

2. It provides explicit control over forgetting via the \eps_{th} parameter.


Weaknesses:

1. GPM performs fairly close to or marginally better than HAT and EWC on most datasets.

2. GPM trades-off between ACC and BWT metrics by strictly controlling forgetting. Hence, it may not be able to identify tasks with very similar structure (or repeated tasks) and may not re-use the Core Gradient Space from previous tasks to improve performance on them (BWT). So, while it minimizes negative BWT, it also restricts positive BWT.


Potential improvements and clarifications:

1. The authors have repeatedly emphasized that their method performs task-free inference. However, doesn't this just mean that the tasks used are such that the inputs encode the task identity and do not require a separate task identifier. Further, on page 6, under Network Architectures, the authors state that apart from permuted MNIST tasks, they use the multi-head setting, i.e., each task has a separate classifier. How does this allow task-identifier free inference?

2. All models have been trained using plain SGD. Would it be possible to extend the method to other optimizers, e.g., Adam?

3. 3 runs are too few to average over. These are supervised learning experiments, so it should definitely be possible to use more runs (at least 5, but preferably 10).

4. At first glance, it seems that the method may have scalability issues since it relies on performing SVD. However figure 2 counter-intuitively shows GPM to be both fast and memory efficient. Is this because these graphs in figure 2 are per-epoch? What about in between tasks when GPM needs to run SVD on all layers? Some more explanation and results are required to better understand how GPM achieves computational efficiency despite relying heavily on SVD per layer after every task.

5. Lastly, no experiments with a single class per task have been performed. This setting is known to be quite challenging in general and induces significantly more catastrophic forgetting (see Kamra et al, 2017). In this setting it is also generally harder to just set batch-norm parameters using just the first task.

[Kamra et al, 2017] Deep Generative Dual Memory Network for Continual Learning. Nitin Kamra, Umang Gupta and Yan Liu. ArXiv 2017.

Please respond to the 5 potential improvements and clarifications mentioned above. I will be happy to raise the score further if the above concerns are addressed.

-------- Post-rebuttal edit -------

The authors have answered all my questions satisfactorily and provided additional experiments wherever it was required including settings where GPM may not outperform other baselines. I believe that the paper is strong and makes a significant technical contribution to the field of CL. Hence, I recommend acceptance and I am updating my score to reflect the same.

---

> ### Author Response · Authors · 2020-11-18
> **Author's Response to Reviewer 4 (2/2)**
>
> **Reviewer’s Comment 4:**  *“At first glance, it seems that the method may have scalability issues since it relies on performing SVD... Some more explanation and results are required to better understand how GPM achieves computational efficiency despite relying heavily on SVD per layer after every task.”* \
> **Response:** Time reported in Figure 2(b) is per-epoch training time for all the algorithms. Figure 2(a,c,d,e) shows relative memory usage in either raw data storage or gradient base storage for different memory-based methods after learning all the tasks. In the per-epoch time comparison, time spent in memory management/update for both our method and the baselines is not included. To address the reviewer’s concern regarding the scalability of our algorithm, we report the training time of our algorithm and compare it with the baselines in Table 2 of the revised manuscript. For PMNIST and CIFAR-100 our algorithm trains faster than all the other baselines while spending only 0.2%, and 3% of its total training time in GPM update (using SVD) respectively. Even in 5-dataset experiment, where ResNet18 is used, our algorithm runs faster than others using only 6% of the training time in GPM update. \
> Although we perform SVD at each layer for GPM update, this is done only once per task. Our formulation uses GPM bases and projection matrices ($MM^T$) of reasonable dimensions (see appendix section C.8). Precomputation of these bases and the projection matrices at the start of each task leads to faster per-epoch training in GPU. This gain in time essentially compensates for the extra time required for GPM update through SVD.  Therefore, our method despite relying on SVD computation is fast and scalable for deeper networks. We have added a summary of this discussion in the revised manuscript.
>
> **Reviewer’s Comment 5:**  *“Lastly, no experiments with a single class per task have been performed….. In this setting it is also generally harder to just set batch-norm parameters using just the first task.”* \
> **Response:** The experiment suggested by the reviewer requires a different evaluation protocol than those used in our paper. This is an instance of class-incremental learning (iCaRL, CVPR’17) where disjoint classes are learned one by one and classification is performed within all the learned classes without task-hint. This scenario is very challenging and often infeasible for the regularization (HAT, EWC etc.) and expansion-based (DEN, APD etc.) methods which do not use old data replay. Using the experimental setting from the paper [1] referred by the reviewer, we implemented the ‘Digit dataset’ experiment where a single class of MNIST digit is learned per task.  Results are the following:
>
> **Result Format -- Method : ACC% (BWT)** \
> EWC : 10% ( -1.00)  ||  DGR : 59.6% ( -0.43)  || DGDMN : 81.8% (-0.15) || GPM : 70.7% (-0.26)
>
> While EWC forgets catastrophically, we perform better than DGR [2], which employs data replay through old data generation. DGDMN[1], an improved data replay method, outperforms all.  In this setup, we believe subset of old data replay from storage or via generation may be inevitable for attaining best performance with minimal forgetting (iCaRL, [3]). In that quest, a hybrid approach such as combining GPM with small data replay would be an interesting direction for future exploration.
>
> [1] Deep Generative Dual Memory Network for Continual Learning. Nitin Kamra, Umang Gupta and Yan Liu. ArXiv 2017\
> [2] Shin, Hanul, Lee, Jung Kwon, Kim, Jaehong, and Kim, Jiwon. Continual learning with deep generative replay. In Advances in Neural Information Processing Systems, pp. 2994–3003, 2017\
> [3] J. Rajasegaran, M. Hayat, S. H. Khan, F. S. Khan, and L. Shao. Random path selection for continual learning. In Advances in Neural Information Processing Systems, pages 12648–12658, 2019

---

> > ### Comment · AnonReviewer4 · 2020-11-24
> > **Thank you for the clarifications**
> >
> > Thank you for providing the clarifications and additional results. You have answered all my concerns satisfactorily and I will be happy to update my score (will update after a discussion with other reviewers during the upcoming discussion period).
> >
> > One final comment though: I noticed that the comparison of GPM with other baselines in the class-incremental learning setup has not been added to the paper. I believe it is important to highlight both strengths and shortcomings of our proposed approaches as researchers. It is completely fine that GPM does not outperform DGDMN in this setting. The experiments are still interesting and insightful in their own right; further they open directions for future research. I would strongly recommend adding these experiments to the final draft (perhaps in the appendix if there is not enough space in the main paper) and discussing the conclusion: "a hybrid approach such as combining GPM with small data replay would be an interesting direction for future exploration" in more detail as a potential direction for future exploration.

---

> > > ### Author Response · Authors · 2020-11-24
> > > **Thank you**
> > >
> > > Thank you for your comments and insightful suggestions. Due to space constraints, we have added the class-incremental learning experiment in the appendix  (section D3) with a brief discussion in the main paper (section 7).  Please let us know if you have further comments.
> > >
> > > Again, we sincerely thank you for your feedback, which has improved the clarity and quality of our submission.

---

> ### Author Response · Authors · 2020-11-18
> **Author's Response to Reviewer 4 (1/2)**
>
> We’d like to thank the reviewer for the detailed and constructive comments on our paper. We provide our point-by-point responses below. Following suggestions from the reviewers, we have revised our manuscript, where changes are marked in blue.
>
> **Reviewer’s Comment 1:**  *“The authors have repeatedly emphasized that their method performs task-free inference…. How does this allow task-identifier free inference?”* \
> **Response:** We thank the reviewer for this comment which gives us an opportunity for clarification. Following GEM, we used single-head evaluation protocol for PMNIST experiment where all the tasks share the same output space. In this setting, the task hint is not necessary. Whereas, for all the other experiments (following HAT, ER_Res and GEM) we used a multi-head evaluation protocol where inference is performed by selecting an appropriate classifier head with task-hint. We made the statement - our method can perform task-identifier free inference - to highlight the findings from the PMNIST experiment. However, we acknowledge that our statement gives a general feeling that in all the experiments we are using task-identifier free inference, which is not the case. Hence, we remove these statements in our revised manuscript. We show in the PMNIST experiment section how our method performs task-hint free inference in that experimental/evaluation setup.
>
> **Reviewer’s Comment 2:**  *“All models have been trained using plain SGD. Would it be possible to extend the method to other optimizers, e.g., Adam?”* \
> **Response:** Our method is a constrained stochastic gradient descent algorithm specialized for continual learning, where we put constraints on the gradients such that weights are updated in the orthogonal direction of the data from the past tasks. We believe our gradient projection scheme can be combined with other optimization algorithms. For example, Adam modifies the original gradient directions, g to $\hat{g}$ through bias-corrected moments ($\hat{m},\hat{v}$). Before updating the weights with this modified gradient, an extra projection step (through equation (6,7) in GPM) can be added to ensure the required orthogonality constraints. The proposed steps are the following: \
> Adam: $\hat{g} = \frac{\hat{m}}{\sqrt{\hat{v}}+\epsilon}$ \
> GPM: $\hat{g}$ = PROJECT ($\hat{g}, \mathcal{M}$) \
> Update : $W = W -\alpha\*\hat{g}$ \
> With this modification to Adam we ran the 10-tasks CIFAR-100 experiment (with lr 5e-3) reported in our paper. We obtained ACC of 70.85% with very small BWT of -0.01. Though the performance is slightly lower than plain SGD, such modification to Adam is still very effective in mitigating catastrophic forgetting.
>
> **Reviewer’s Comment 3:**  *“3 runs are too few to average over. These are supervised learning experiments, so it should definitely be possible to use more runs (at least 5, but preferably 10).”* \
> **Response:** Following reviewer’s suggestion we have updated our results averaging over 5 runs.

---

### Official Review · AnonReviewer3 · 2020-10-29
**Nice idea**

**Rating:** 8
**Confidence:** 5

**Review:**

I found the idea quite novel. Lately in continual learning the focus has been more on NAS type ideas and algorithms, but this work is a nice divergence from this direction. The idea of optimizing in a space orthogonal to the previous task is novel. The execution of the idea is nothing special since it's using standard linear algebra, but I gave the authors full merit to the idea itself.

My higher score is mostly due to the fact that the experiments are limited. The benchmark algorithms definitely miss some recent works from 2019 and 2020. They should be included as otherwise the superior performance of the algorithms is questionable.

See for example:
https://arxiv.org/abs/2006.04027
Ju Xu and Zhanxing Zhu. Reinforced continual learning. In Advances in Neural Information Processing Systems, pages 899–908, 2018

---

> ### Author Response · Authors · 2020-11-18
> **Author's Response to Reviewer 3**
>
> We thank the reviewer for the review and comments on our paper. We are happy to hear that he/she finds our idea interesting. We address the reviewer’s comment below:
>
> **Reviewer’s Comment:**  *“The benchmark algorithms definitely miss some recent works from 2019 and 2020. They should be included as otherwise the superior performance of the algorithms is questionable.”*\
> **Response**: In our paper, we have already compared our method with the state-of-the-art (SOTA) regularization and memory-based methods. As per reviewer’s suggestion, we have added a new experiment to compare with some of the representative works from network expansion-based continual learning methods which include Reinforced Continual Learning (RCL)[1] and recently proposed SOTA method – Adaptive Parameter Decomposition (APD)[2].   In this experiment, 20 sequential tasks are learned from CIFAR-100 Superclass dataset [2], where each task contains 5 different but semantically related classes. Results are given in Table 3 of the revised manuscript with necessary explanations. Brief summary of the results is given below:
>
> **Methods/Metric | ACC (%) | Network Capacity (%)** \
> RCL | 51.99% | 184%\
> APD | 56.81% | 130%\
> GPM | 57.72% | 100%
>
> Our method (GPM) outperforms all the other methods while using significantly less network capacity.  This implies GPM encourages higher sharing among tasks thus makes efficient usage of given network capacity. \
> Following suggestions from the reviewers, we have revised our manuscript, where changes are marked in blue.
>
> [1] Ju Xu and Zhanxing Zhu. Reinforced continual learning. In Advances in Neural Information Processing Systems, pages 899–908, 2018\
> [2] Jaehong Yoon, Saehoon Kim, Eunho Yang, Sung Ju Hwang. Scalable and Order-robust Continual Learning with Additive Parameter Decomposition. ICLR, 2020.

---

### Author Response · Authors · 2020-11-19
**To all the Reviewers**

We thank all the reviewers for their valuable suggestions and comments. We have made the following changes to our revised manuscript as per their suggestions:
1. Added training time comparison in section 7
2. Added a new experiment for comparison with expansion-based methods in section 7
3. Updated the results (in Table 1, 8, 9) for 5 runs.
4. Clarified in which experiment we use task-hint and where we don’t
5. Added relevant references in related works
6. Added miscellaneous explanations/discussions
7. Added class-incremental learning experiment in appendix section D.3

**Correction** - We used ResNet18 for 5-dataset experiment. While this was correctly specified in the appendix (C2), in the main text (section 6) we wrote that we use AlexNet for that experiment. We have corrected this mistake in the revised manuscript.

All changes are marked in blue in the revised manuscript.

Please let us know your comments on our response and revised manuscript. We are open to further discussion if needed.

---

### Comment · ~Yeming_Wen1 · 2021-05-06
**Some comments on comparison with expansion-based methods**

Hi,

I really like the idea of this paper, it has a solid contribution to the continual learning community. Congrats on the oral acceptance!

I have a comment on the section where you compare to expansion-based methods. I wonder how does table 3 looks if we change the task to simple SPLIT-CIFAR-100, the same setup in the GEM paper, Lopez-Paz & Ranzato (NeurIPS 2017). Does GPM still outperform PGN a  lot in this continual learning setup?

I asked this because I found PGN can reach really competitive accuracy, figure 3b in the BatchEnsemble paper, Wen et al. (ICLR 2020). Because of this, we think the expansion-based methods is more promising than the memory-based methods, leading to the idea of using rank-1 perturbation weight to reduce the memory cost of PGN.

Thanks

---

### Decision · Program_Chairs · 2021-01-07
**Final Decision**

**Decision:**

Accept (Oral)

**Comment:**

The paper proposes a new approach to continual learning with known task boundaries that is scalable and highly performant, while preserving data privacy.  To mitigate forgetting the proposed approach restricts gradient updates to fall in the orthogonal direction to the gradient space that are important for the past tasks. The main novelty of the approach is to estimate these subspaces by analysing the activations for the inputs linked for each given task.

All reviewers give accepting scores. R2, R3 and R4 strongly recommend accepting the paper, while R1 considers it borderline.

The authors provided an extensive response carefully considering all reviewers' comments. New experiments were introduced (training time analysis and comparisons with expansion-based methods), and several clarifications were added.

All reviewers agree that the paper is well written and its literature review adequate.

The main concern of R1 was the similarities with OGD (Farajtabar et al. 2020). R1 considered the authors’ response acceptable. R2, R3 and R4 consider the contribution well motivated and significant and highlight its simplicity. The AC agrees with this assessment.

The empirical evaluation covers most of the typical benchmarks in CL. Very strong results are reported on a variety of tasks both in terms of performance and memory efficiency, as agreed by R2, R3 and R4.

Overall the paper makes a strong contribution to the field of CL.